# A Bibliometric Analysis of Groundwater Access and Its Management: Making the Invisible Visible

Priyanka Lal [1] , Biswaranjan Behera [2], Malu Ram Yadav [3] , Eshita Sharma [4] , Muhammad Ahsan Altaf [5] ,
Abhijit Dey [6] , Awadhesh Kumar [7] , Rahul Kumar Tiwari [8] , Milan Kumar Lal [8,\*] and Ravinder Kumar [8,\*]

1   Department of Agricultural Economics and Extension, Lovely Professional University, Jalandhar GT Road (NH1), Phagwara 144002, Punjab, India
2   ICAR-Indian Institute of Water Management, Bhubaneswar 751023, Odisha, India
3   Division of Agronomy, Rajasthan Agricultural Research Institute, Sri Karan Narendra Agriculture University, Jaipur 303329, Rajasthan, India
4   Department of Molecular Biology and Biochemistry, Guru Nanak Dev University, Amritsar 143005, Punjab, India
5   College of Horticulture, Hainan University, Haikou 570228, China
6   Department of Life Sciences, Presidency University, 86/1 College Street, Kolkata 700073, West Bengal, India
7   ICAR-National Rice Research Institute, Cuttack 753006, Odisha, India
8   ICAR-Central Potato Research Institute, Shimla 171001, Himachal Pradesh, India
\*   Correspondence: milan.lal@icar.gov.in (M.K.L.); chauhanravinder97@gmail.com (R.K.)

**Abstract:** The sustainable management of groundwater resources is required to avoid a water crisis. The current study focused on a bibliometric analysis of groundwater access and management to assess research progress. The study was based on data from Dimensions.ai generated using the search terms "Groundwater", "access", and "management" for the period from 1985 to 2022. A total of 534 documents were identified as relevant and retrieved in CSV format. The intellectual structure of the retrieved data was visualized and analyzed using VoS viewer software (version 1.6.18). The analysis showed that the field of earth sciences had the highest number of publications on groundwater access and management (358), followed by the environmental sciences (155). Most of the articles (267) were about Sustainable Development Goal 6, which focuses on ensuring access to clean water and sanitation. The co-authorship analysis for the countries indicated that the United States has the most impact and research, and all other countries have established clusters around it. The citation analysis of the organizations showed that the International Water Management Institute, Charles Sturt University, and Wageningen University and Research were the top three organizations in terms of total citations (825, 611, and 584, respectively), indicating the most effect. The citation analysis for the sources indicated that the "Water" journal had a greater impact on readers with respect to groundwater research. Numerous parties are involved in the groundwater investigation; hence, a broad multidisciplinary approach is required. Therefore, researchers should work together rather than alone to address the problem of sustainable groundwater management.

**Keywords:** water resource; bibliometrics; bioinformatics; groundwater; water management

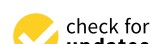



## 1. Introduction

Groundwater supplies drinking water to at least half of the world's population, and over 2.5 billion people worldwide rely primarily on groundwater resources to meet their basic daily water demands [1]. Between 70 and 90 percent of domestic water consumers reside in developing sub-Saharan nations [2]. Similarly, in the United States, nearly 38% of the population uses public groundwater sources for drinking water. Nearly 79% of the people in Southeast Asian and Pacific nations use a drinking water supply in the form of groundwater [2]. It is crucial to understand the reliance on this resource and the amount and quality consumed due to the widespread use of groundwater in agricultural practices

and industrial processes. However, the worldwide trend of groundwater depletion shows that groundwater is increasingly threatened due to excessive utilization [3].

Groundwater resources are now in danger due to overuse, but the increasing reliance on them has also contributed to their depletion. Concerningly, groundwater is being depleted on a global scale despite its importance for both industrial and agricultural processes. The depletion of groundwater resources caused by the use of groundwater for irrigation, particularly in agricultural areas, has made it more challenging for people to acquire the water they require for their everyday activities. In addition to the depletion threat, groundwater quality is also at risk [1]. Groundwater is vulnerable to contamination from human activities, such as using pesticides and fertilizers in agriculture and releasing toxic chemicals into the environment through industrial processes. This contamination can lead to serious health problems, particularly in communities that rely on groundwater for their drinking water supply [4].

Today, groundwater access and management are the primary focuses for which many strategies have been employed. Access must be prompt and supervised by a higher authority for adequate extraction. Water access through aquifers has its restrictions to avoid negative consequences. Groundwater management must meet numerous expectations from consumers, scientists, farmers, and others. Thus, authors [5,6] have proposed methods for extracting water from certain aquifers in a sustainable manner. Researchers have sought to determine how much water should be extracted in order to have the least impact on the ecosystems [7]. The individuals and entities involved in accessing and extracting groundwater have driven both planned and unplanned development, potentially exacerbating the stress on this vital resource. Their decisions, centered on individual interests, have significant long-term impacts, including a decrease in groundwater levels, the desaturation of aquifers, rising energy requirements to pump water from deeper depths, and a decline in water quality due to saltwater intrusion in coastal regions. On the other hand, there are parts of countries where groundwater development is still low-key, despite the availability of sufficient resources; similarly, canal command areas suffer from water logging and soil salinity as groundwater levels gradually rise. As a result, a comprehensive strategic road map is required to sustain groundwater sources and avoid a water crisis, thereby making the unseen apparent [8,9]. In this context, the current study focused on a bibliometric analysis of groundwater access and management to determine how far we have progressed.

It is essential to exercise proper management of and conservation efforts for this unique resource to reduce the risk of groundwater depletion and contamination. This includes applying sustainable practices in agriculture and industry, such as minimizing water use and following best management practices to prevent the discharge of pollutants into the environment [4]. In addition, this includes reducing the amount of energy used in agriculture and industry. Implementing rules and regulations aimed at preserving and maintaining this resource also falls under the jurisdiction of governments and other organisations, both of which play important parts in encouraging the sustainable use and management of groundwater resources [10].

The issue of groundwater quality deterioration may lead to significant and possibly irreversible impacts on human health and ecological services. On the one hand, groundwater resources are the final receivers of pollutants in the process of the water cycle. On the other hand, it is very difficult to restore them after they have been polluted. Given these two facts, it is possible that the issue of groundwater quality deterioration may lead to these effects [11]. In addition, the pollution of groundwater drastically cuts down on the amount of groundwater resources that can be extracted, which ultimately results in a severe lack of available water. As a result, there is an immediate and pressing requirement to successfully reduce groundwater hazards on a global scale through eradicating groundwater contamination, improving groundwater resilience, and implementing sustainable groundwater management [12,13]. The measurement of groundwater levels is an essential component of agricultural water management because it enables the precise and straight-

forward forecasting of groundwater levels on farms. Consequently, this part of agricultural water management is often overlooked. The ever-increasing demand for groundwater over the past several decades has brought to light the critical nature of accurate evaluations and projections. Researchers have combined machine learning with mathematical modelling and geospatial analysis in order to forecast changes in groundwater levels and determine the state of groundwater, including chemical concentrations. This has allowed researchers to determine the state of groundwater. Previous research has demonstrated that the use of the aforementioned strategies and techniques is successful [8–20].

Recently, experts have focused more on how to attain the Sustainable Development Goals. The Sustainable Development Goals (SDGs) are a series of 17 goals that have been established by the United Nations in an effort to put an end to poverty, safeguard the environment, and secure prosperity for all people. A universal, equitable, and cheap supply of potable water is one of the most important aims of the Sustainable Development Goals (SDGs) (SDG 6). Because it is a primary supply of potable water for many communities all over the world, groundwater is an extremely important component in the process of accomplishing this objective [21]. Access to clean and reliable groundwater is still difficult to come by in many parts of the world, despite the significance of this resource. It is common for the resources that contain groundwater to be contaminated, which results in hazardous drinking water and possible health problems for the communities who are affected. In addition, many different communities, particularly those located in rural areas, do not have access to contemporary infrastructure for the extraction and transport of groundwater. As a consequence of this, a sizeable percentage of the population must rely on unreliable sources of water, such as surface water that has been polluted [1,3,22].

It is vital to apply groundwater management strategies that are both successful and sustainable if one wishes to ensure that access to safe groundwater will continue to improve and be maintained. This comprises the development of appropriate extraction and distribution systems, regular monitoring and assessment of groundwater quality, and preservation of groundwater resources from contamination. Access to safe and reliable groundwater is vital to meeting the Sustainable Development Goals (SDGs), notably SDG 6 [21]. The empowerment of communities to better their health, livelihoods, and overall well-being can be achieved by implementing efficient groundwater management policies and guaranteeing safe access to this essential resource.

The present evaluation will shed light on the key areas and how far we have progressed regarding groundwater toward meeting SDG Goal 6. In this regard, we have formulated the following research questions, which we have attempted to address in this study:

(a) What bibliometric analysis methodology should be used?
(b) What is the relationship between the publication and citation growth of groundwater access and management?
(c) What are the total numbers of publications in each research category and what is the most prevalent field of such articles?
(d) What is the number of publications associated with a specific SDG?
(e) What is the intellectual structure that depicts prolific writers and leading institutes performing groundwater access and management research?
(f) How has the literature evolved?
(g) What is the impact of collaboration, and which countries have the most collaborations?
(h) Using the authors' keywords and themes, what are the conceptual dynamics of groundwater research in India?

This study conducted a thorough bibliometric analysis [7] which aided in the systematic assessment of global advances [8,9,16]. Essentially, it employed a methodical approach to comprehend the scientific and intellectual structures via quantitative approaches such as authors, citations, organizations, sources, keywords, etc. The process became more apparent as information was collected in a specific field [10,17–19]. The bibliometric analysis provided the researchers with information to investigate a topic of scientific interest [20]. With this in mind, this study aimed to understand the literature-based relationship between

groundwater availability and its management. This paper is divided into sections. The first section describes the research technique, which included data gathering, data mining, and data cleaning, and it is followed by a section that analyses the results and presents them.

## 2. Materials and Methods

### 2.1. The Data and Methodology

The current study was a bibliometric investigation of groundwater access and management. Groos and Pritchard [23] invented the phrase "bibliometric analysis", which refers to a collection of quantitative tools for tracking and analyzing the flow of literature on a specific issue [24]. We used the steps shown in Figure 1 for this investigation, and the study period was from 1985 to 2022.

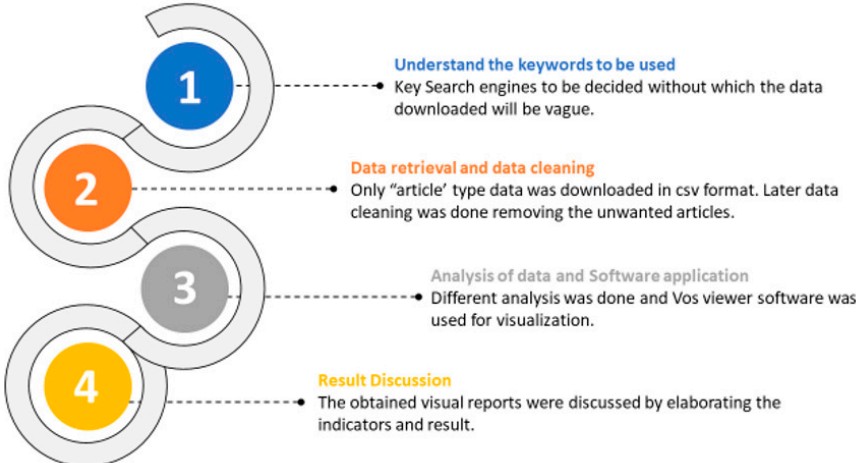

**Figure 1.** Steps in a bibliometric analysis.

The research was based on data from Dimensions.ai, an interlined research information system offered by Digital Science. Previously, only the Web of Science (WoS) published standard and homogeneous research publications [25]. Later, the Scopus database and Google Scholar emerged as competitors. Given the financial constraints for subscriptions faced by curious individual researchers in developing and underdeveloped countries, Google Scholar was the preferred option. The platform enables stakeholders to simply obtain data and construct concepts. After COVID-19, the pace of bibliometric studies increased since the crisis revealed a different set of needs for analyzing trends in real-time bibliometrics [26]. Dimensions enables scholars to search full-text data for various years, as well as scholarly works such as pre-prints, articles, chapters, conferences, monographs, and edited books. This expanded data collection for examination opens up new research opportunities. Researchers use machine learning techniques to construct links between items, while data upgrades such as per-object categorizations and person and institution disambiguation provide further context. However, its scientific validity and applicability for bibliometric investigations are controversial [27,28].

In this study, the Dimensions database was used to gather information on authors, article titles, author affiliations, and other relevant details, making it a suitable source for bibliometric research. The data obtained offered various analytical functions, including citation analysis and subject analysis. There were no language restrictions, and all data extracted was in English as the search query required English titles and abstracts. This enabled the researchers to assess the content of non-English documents through their titles and abstracts. The success of a bibliometric study largely depends on the accuracy of the search query. Hence, the search query was carefully crafted based on a comprehensive review of existing scientific papers and the identification of research gaps. The aim was to ensure that the search query was comprehensive and produced high-quality and reliable results.

*2.2. Database Selection and Search Querry*

The search engine query employed for the analysis included all relevant types of terms that served our study's theme. The search query should be more particular rather than generic, as generic queries may result in inaccurate data. "Groundwater", "access", and "management" were the search terms utilized in the study. The search was conducted for titles and abstracts, and it spanned the years 1985 through 2022. Documents from unpublished sources, pre-prints, book chapters, etc. were all excluded. Data were cleared of unwanted articles after obtaining them from Dimensions.ai, which had confined the search to only article-type publications. Following data filtering, a total of 534 documents were identified as relevant and retrieved in CSV format.

The following step analyzed the retrieved data in VoS viewer software (version 1.6.18). The program's use is rapidly increasing in various bibliometric investigations for intellectual structure visualization and analysis [29,30]. Aside from that, data trend analysis was performed.

*2.3. Data Extraction and Analysis*

The data were collected in MS Excel CSV format for further analysis. Excel software was used to perform tabular analysis while VOS Viewer software (version 1.6.18) was utilized for visualizing maps [31]. Citation analysis was employed to evaluate the scientific significance and impacts of the publications. A "connection strength" metric was derived from the visualized maps to assess international research collaboration among the participating countries. This metric measures the degree of scientific collaboration between any two countries and is depicted by the thickness of the connecting lines between them. The thicker the line, the greater the collaboration and, therefore, the higher the importance of the connection strength.

## 3. Results and Discussion

*3.1. Number of Publications*

Five hundred and thirty-four documents were carefully examined after the data that were pertinent to the topic of the study were taken into consideration. The total number of articles that met each criterion for the broad subject area of study is displayed in Figure 2. Based on the findings of the investigation, it was discovered that the field of earth sciences had the highest number of publications on the topic of groundwater access and management (358), followed by the field of environmental sciences (155). We also analyzed the total number of publications published on each criterion of the Sustainable Development Goals (SDGs) as groundwater availability and management play significant roles in accomplishing these goals. Figure 3 depicts the outcomes of the study as they were obtained. The usage of groundwater, as well as its management and long-term viability, were the topics of these publications. From Figure 3, we can deduce that most of the articles were completed for Sustainable Development Goal 6, which focuses on ensuring access to clean water and sanitation (267 numbers). All publications pertaining to water were read with equal attention to groundwater. As a result, groundwater's status as a fallback resource necessitated the application of focused and specific attention.

The total number of publications during the study period is presented in Figure 4. The figure depicts that there was an increasing trend of such publications from 2000 to 2021, with a slight dip in the years 2019–2020. Sustainable groundwater management is a crucial global issue and is now one of the main focuses in recent keyword analyses. The United Nations Development Programme has linked over 50 specific targets in the Sustainable Development Goals (SDGs) to groundwater, with particular emphasis on SDG 6 (clean water and sanitation), SDG 12 (responsible consumption and production), and SDG 13 (climate action). These goals not only address water-related issues such as water quantity and quality, but they also consider the economic, social, and environmental implications of groundwater management [4].

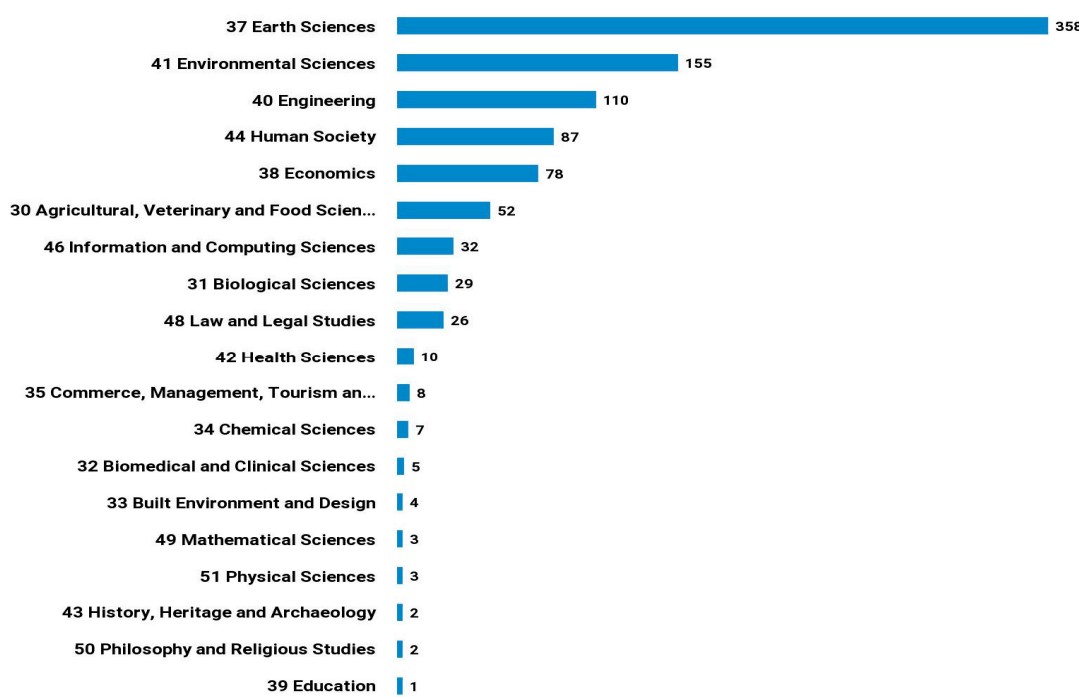

**Figure 2.** Number of publications in each research category.

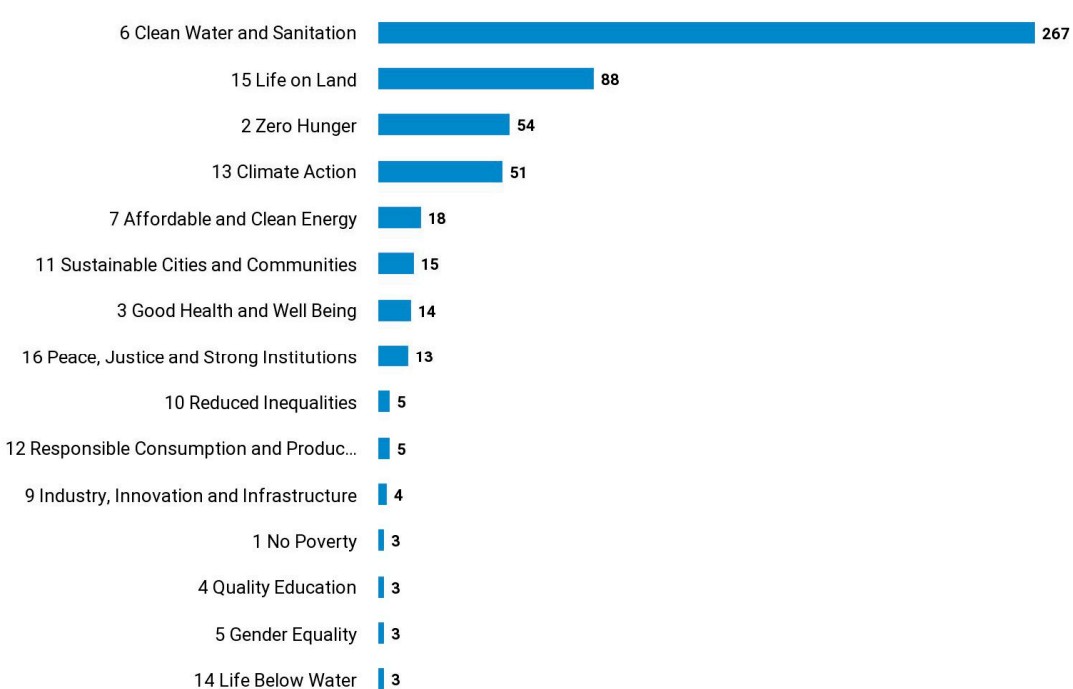

**Figure 3.** Number of publications on the SDG themes.

However, current research on groundwater vulnerability assessments (GVAs) has primarily focused on the water aspect and neglected the other aspects of the SDGs. It is important to note that GVA is merely a starting point in sustainable groundwater management, and further work is required to achieve the SDGs. This includes hydrological surveys, land planning, water allocation and use, water pollution control and monitoring, and health risk assessments, among others [12]. To bridge the gap between GVAs and the SDGs in groundwater management, it is proposed to integrate GV more closely with the SDGs. This will provide a more comprehensive and effective approach for achieving groundwater sustainability in future research [12,15,21].

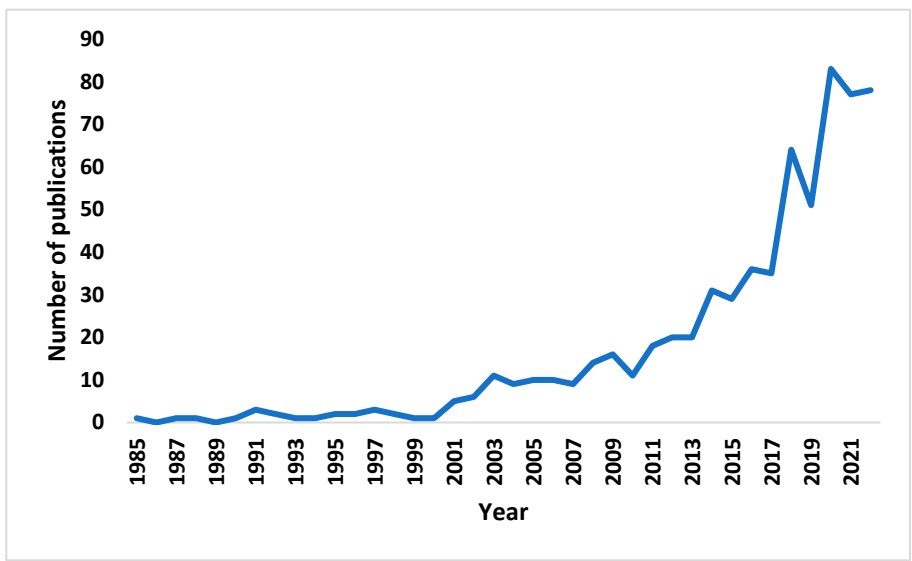

**Figure 4.** Growth of publications during the study period.

*3.2. Performance Analysis*

3.2.1. Top Author Analysis

The number of documents and citations a certain author has contributed to is one of the most important factors in determining that author's impact. Our study findings are presented in tabular and graphical formats, respectively. Table 1 features the prolific authors, along with the number of published documents and the average number of citations per document. The degree of collaboration between individuals can be identified and quantified by analyzing the number of links established through co-authorship.

**Table 1.** Top 15 authors analysis.

| Author's Name | Total Number of Documents | Citations | Citations per Document |
|---|---|---|---|
| Shah, T. | 2 | 155 | 77.50 |
| Maheshwari, B. | 6 | 82 | 13.67 |
| Ward, J. | 3 | 77 | 25.67 |
| Dillon, P. | 2 | 63 | 31.50 |
| Dave, S. | 2 | 59 | 29.50 |
| Kookana, R. | 2 | 59 | 29.50 |
| Faybishenko, B. | 3 | 53 | 17.67 |
| Varadharajan, C. | 3 | 53 | 17.67 |
| Chew, M. | 2 | 50 | 25.00 |
| Chinnasamy, P. | 2 | 45 | 22.50 |
| Agarwal, D. | 2 | 39 | 19.50 |
| Arora, B. | 2 | 39 | 19.50 |
| Park, J. | 2 | 39 | 19.50 |
| Sahu, R. | 3 | 39 | 13.00 |
| Pinto, U. | 2 | 18 | 9.00 |

The analysis of authors given in the table above was based on the top 15 writers found in the analysis. Figure 5 depicts the visualization analysis. To comprehend the co-authorship analysis, VoS software was used. The co-authorship analysis showed the authors' relatedness based on how strong their links were and which important authors collaborated. Because its study involves many stakeholders, groundwater requires a broad multidisciplinary approach. As a result, researchers no longer act independently and collaborate to address the issue.

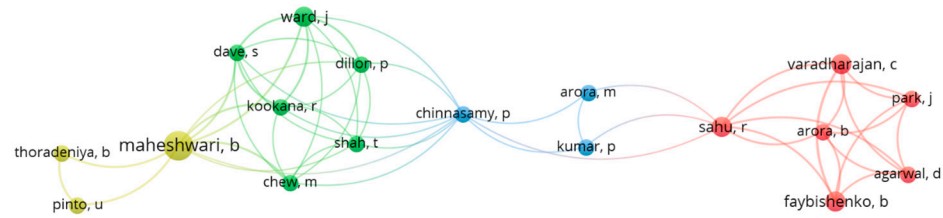

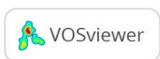

**Figure 5.** Co-authorship analysis of the authors.

Under certain constraints, a co-authorship analysis of authors was performed using VoS viewer. We kept the minimum number of documents per author at two and the maximum number of authors per document at twenty-five. Articles that did not meet these requirements were automatically deleted from the study. Figure 5 depicts the findings of the analysis, which were displayed on the output window of the VoS viewer. There were 180 authors who met the aforementioned requirements, but only 18 of them were related to each other. There were four clusters discovered in total. Each cluster demonstrated how the authors collaborated and shared intellectual space. Similarly, a co-authorship analysis was performed on the countries that collaborated. It is a notion that co-authorship is one of the most measurable forms of scientific collaboration that is well-documented, and the output of these interactions creates a 'co-authorship network'. From Figure 5, the four clusters depict the strong networking among the four groups of authors. Maheshwari, Sahu, Chinnasamy, etc., were among the top authors that contributed to the literature.

### 3.2.2. Co-Authorship Analysis of Countries

The same data obtained were used for a country co-authorship analysis. Basic criteria were applied in order to filter and obtain data on a strong collaboration network among countries. The minimum number of documents per country was kept at four. Therefore, only 42 of the 97 countries met the requirement. Figure 6 depicts the visualized map.

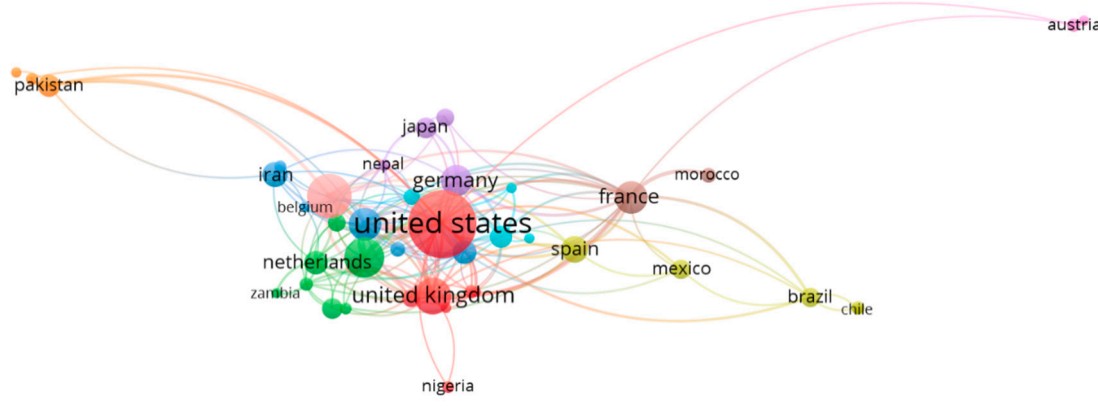

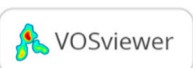

**Figure 6.** Co-authorship analysis of the countries.

Based on the research, it was discovered that ten clusters of countries had developed. The countries are represented on the map by labels and circles. The sizes of the circles represent the amounts of documents produced by each country. The larger the circle and label, the greater the country's impact on a specific study. The United States has the most impact and research, and all other countries have established clusters around it. This indicates a strong network of collaboration among these countries.

### 3.3. Citation Analysis

The information obtained was utilized to complete a citation analysis. It is a popular bibliometric strategy since it uses and examines citations in one paper to develop links with other researchers. Sandison [32] declared that a citation is more than merely a piece of bibliographic data at the conclusion of a text comprised of comments, footnotes, etc. or text retrieved from a citation index. In fact, a citation is the depiction of an author's decision to show the relationship between the text they are producing and the work of another (at a particular point). Similarly, Shaw [33] wrote that "citation develops a relation among authors which is a measure of the amount to which they interact indirectly through the literature". Figures 7 and 8 show the visual map of the citation analysis of the organizations and sources, respectively.

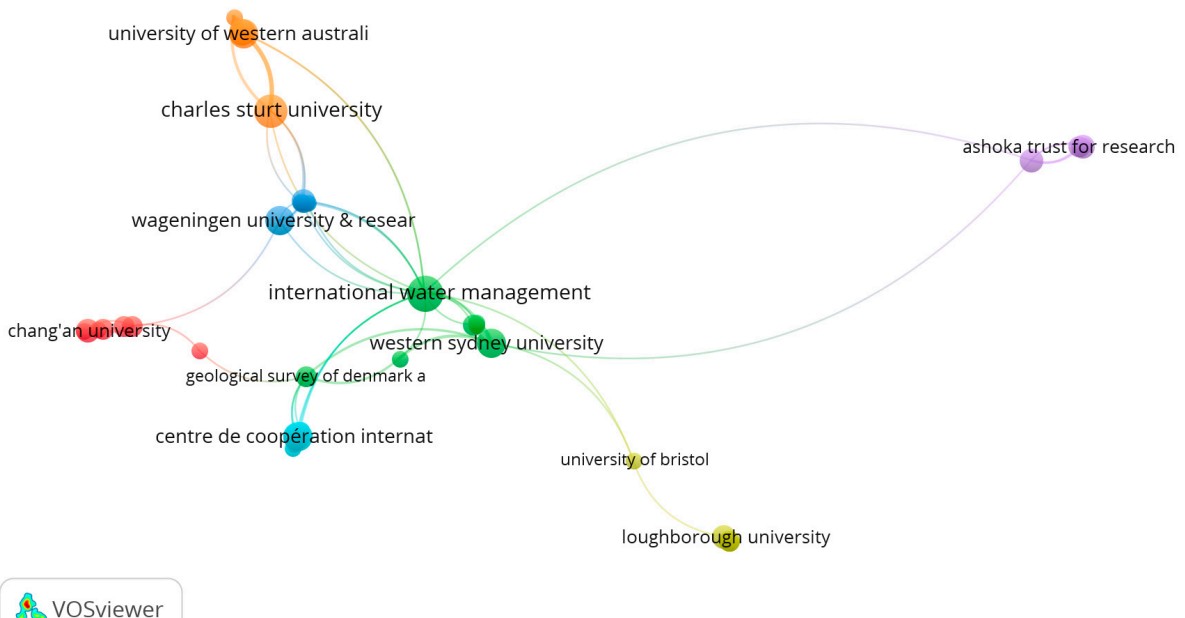

**Figure 7.** Visualization map of the citation analysis (organization).

To understand how organizations cite each other, the minimum number of documents from each organization was set at two. According to the criteria, there were 40 total related organizations and 7 total clusters. The International Water Management Institute, Charles Sturt University, and Wageningen University and Research were the top three organizations in terms of total citations (825, 611, and 584, respectively), indicating the most effect.

According to Egghe and Rousseau [34], the presence of a cited document in a reference list implies that the author believes there is a relationship (e.g., resemblance in the subject, issue, approach, etc.) between the referenced and cited materials. They pointed to citation analysis as a field that studies these links.

Citation analysis for sources was also performed in this regard. The criteria used in this analysis were a minimum of three documents per source. This requirement was met by 47 sources, resulting in 6 clusters. Only 24 of the sources examined in the investigation had a strong relationship. The larger the bubble, the greater the impact of a specific source.

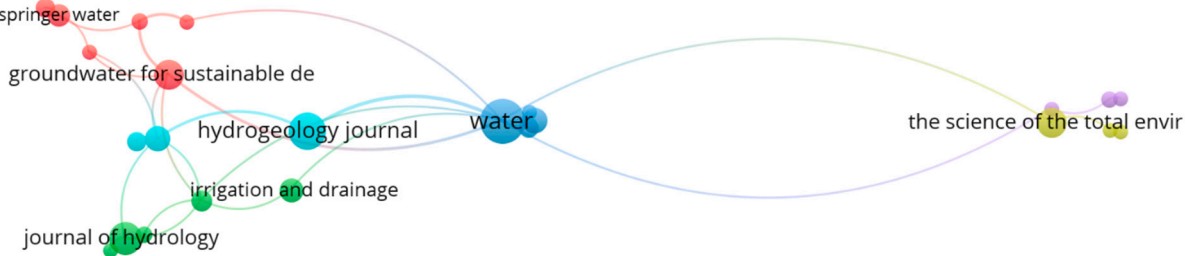

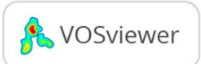

**Figure 8.** Visualization map of the citation analysis (sources).

*3.4. Pillars and Approach of Groundwater Access and Management in India*

Through the review of the literature, we identified important pillars for groundwater access and management (Figure 9). The four pillars are as follows:

1. Understand the extraction: In order to extract groundwater, an optimal extraction procedure should be followed as time should be given for the proper recharge of the groundwater resources.
2. Protect against pollution: Proper government policies need to be formulated to protect groundwater against pollution. Checks should be kept on the release of pollutants to groundwater.
3. Avoid overdraft: The extraction procedure should not be performed beyond equilibrium yield. This will lead to more extraction than recharge of the groundwater.
4. Ensure equal access: The government policy should focus on equal access irrespective of social status or place. No community should be kept aside to use and access the groundwater. Thus, the sustainability of groundwater can be achieved.

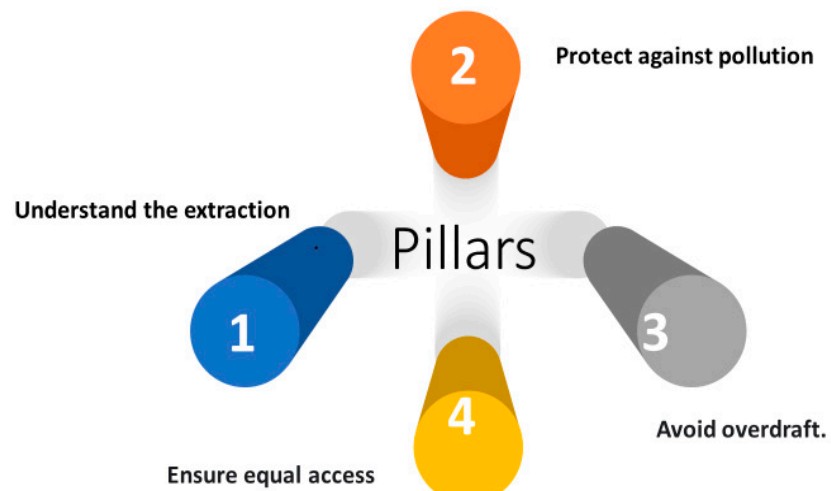

**Figure 9.** Pillars of groundwater access and management.

Sustainability is always concern when we talk about water and its sources. The approaches for groundwater management are shown in Figure 10. The three approaches are necessary for sustainable groundwater management.

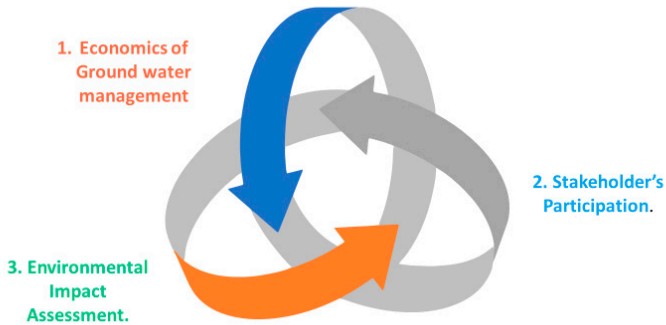

**Figure 10.** Approaches for sustainable groundwater management.

The long-term preservation of groundwater resources is a very important issue that requires the coordination and participation of many different groups. Among these factors are the economics of groundwater management, the involvement of stakeholders, and an assessment of the environmental impact. Groundwater management economics look at how to remove water from the ground in a way that is good for both the economy and the environment. This means thinking about how much different ways of extracting the resource will cost and how they might affect the resource in the long run. On the other hand, stakeholder participation is when the community works together to manage and protect groundwater. This can include education and awareness programs, as well as chances for people in the community to give their thoughts and opinions on how groundwater is managed. The environmental impact assessment is an important tool for figuring out how illegal groundwater extraction and other activities affect groundwater resources. This information can be used to make management plans that work better and last longer. The groundwater resources cannot be maintained without the right combination of these three things. By combining the insights and perspectives of groundwater management economics, stakeholder participation, and environmental impact assessment, it is possible to create effective strategies for sustainable groundwater management.

## 4. Conclusions

The present bibliometric analysis of groundwater access and management concluded that 42 countries had a strong network of collaboration, among which the United States had the most impact and research. A total of 40 organizations in 7 clusters were related, with the International Water Management Institute, Charles Sturt University, and Wageningen University and Research as the top three organizations having the most effect on groundwater access and management research. Among the different sources, the water journal had a greater impact with respect to groundwater research. In conclusion, groundwater is an important source of water for millions of people all over the world. This water supplies these people with the water they require for their day-to-day activities, as well as their survival. However, the extensive reliance on this resource has also contributed to its depletion and contamination. As a result, it is becoming increasingly vital to establish sustainable practices and policies that are aimed at protecting and maintaining this valuable resource. We can only guarantee that future generations will have access to this resource if we make an effort to preserve and safeguard groundwater by taking the appropriate precautions. The study of groundwater involves many stakeholders and, hence, needs a broad multidisciplinary approach. Therefore, researchers should collaborate to address the issue, and they should no longer act independently in sustainable groundwater management.

**Author Contributions:** Conceptualization and design, P.L., M.K.L., R.K.T. and A.K.; interpretation of data and original draft preparation, M.R.Y., B.B., E.S. and M.A.A.; analysis and interpretation of data and writing, P.L., R.K. and E.S.; performed analysis and interpretation of data and resources, M.R.Y., M.A.A. and A.D.; revised draft manuscript and performed supervision, R.K. and A.K. All authors have read and agreed to the published version of the manuscript.

**Funding:** This research received no external funding.

**Institutional Review Board Statement:** Not applicable.

**Informed Consent Statement:** Not applicable.

**Data Availability Statement:** All data generated or analyzed during this study are included in this published article.

**Acknowledgments:** The authors would like to acknowledge the Director of the ICAR-Central Potato Research Institute, Shimla, for their constant support and guidance.

**Conflicts of Interest:** The authors declare that they have no known competing financial interest or personal relationships that could have appeared to influence the work reported in this paper.

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
