# Peer review of "A Bibliometric Analysis of Groundwater Access and Its Management: Making the Invisible Visible"

_water, doi:10.3390/w15040806_

Round 1
Reviewer 1 Report
General Comments:
1. Title = A Bibliometric review on groundwater access and its management-making the invisible visible.
In this way the word “making” refers to management and I believe that confuses the reader. I suggest the “-“ symbol to be replaced with “:”.
2. The paper needs to be crosschecked to correct grammar/syntax errors. It must be reviewed by an efficient (preferably native) English-speaking reviewer before being reconsidered for publication.
3. A Graphical Abstract is needed.
4. References’ presentation in the text is not the correct. According to “Instruction for Authors” àReferences must be numbered in order of appearance in the text (including table captions and figure legends) and listed individually at the end of the manuscript.
Specific Comments:
· Line 35: “……indicated the water journal with a larger bubble had a greater impact with…”. I think that the phrase “with a larger bubble” is not necessary here in the Abstract, as the reader does not know the methodological tools for results evaluation yet. It is an information that concerns results discussion.
· Line 74-75: “Researchers have sought to determine how much water should be extracted in order to have the least impact on the ecosystem (Koroneos et al., 2013).” The place of this sentence at this paragraph is not the appropriate because before this sentence authors presents the scope of the study as follow “In this context, the current study focuses on a bibliometric analysis of groundwater access and management to determine how far we have 72 progressed.” And after the sentence “Researchers have sought to determine how much water should be extracted in order to have the least impact on the ecosystem (Koroneos et al., 2013).” An extra presentation is occurred with more information. This interrupts the presentation of the study. So, the sentence “Researchers have sought to determine how much water should be extracted in order to have the least impact on the ecosystem (Koroneos et al., 2013).” ought to be moved.
· Line 103: I believe that in the text there is no answer to the question h) Using the author's keywords and themes, what are the conceptual dynamics of groundwater research in India? Could you please mention it (if I miss something) or add it?
· Lines 126-127: Did the authors try searches with other terms so as to gain higher number of results? e.g. instead of “access” to use “exploitation” or “extraction”.
· Line 128: “…..and it spanned the years 1989 through 2022.” The authors declared in the abstract that the period is from 1985 to 2022. Which is the correct period?
· Lines 159: I do not believe that the statement “The figure depicts that there was an increasing trend of publications from 1985 to 2021……..” properly described the Figure 3. I believe that the increasing trend starts after 2000.
· Line 212: A dot “.” is needed before the word Sandison as a new sentence must begin.
· Lines 213-214: “…….piece of bibliographic data at the conclusion of a text ä send comments, footnotes, and so on a or retrieved from a citation index.” I do not understand the “ä” and the “a” in the phrase “…..and so on a or retrieved from a citation index”. These two “a” are added by mistake? If there is a reason, please explain.
Author Response
Response to the reviewer
The authors thank the reviewer for reviewing the manuscript thoroughly and providing positive and constructive remarks. The changes made in the manuscript were in track change mode. The response to the reviewer is given in the following points:
Comment 1: Title = A Bibliometric review on groundwater access and its management-making the invisible visible.
In this way the word “making” refers to management and I believe that confuses the reader. I suggest the “-“ symbol to be replaced with “:”.
Response 1: Authors agree to the concern raised by the reviewer. The suggested changes have been incorporated in the title of the manuscript. Now the title of the manuscript is “A Bibliometric analysis on groundwater access and its management: making the invisible visible”
Comment 2: The paper needs to be crosschecked to correct grammar/syntax errors. It must be reviewed by an efficient (preferably native) English-speaking reviewer before being reconsidered for publication.
Response 2: Authors agree to the concern raised by the reviewer. The suggested changes have been incorporated throughout the manuscript. Which can be observed in the track change mode.
Comment 3: A Graphical Abstract is needed.
Response 3: As per the suggestion of the esteemed reviewer, a Graphical Abstract has been incorporated into the manuscript.
Comment 4: References’ presentation in the text is not the correct. According to “Instruction for Authors” à References must be numbered in order of appearance in the text (including table captions and figure legends) and listed individually at the end of the manuscript.
Response 4: Authors agree to the concern of the kind reviewer. Citations and references have been rectified according to the journal guidelines.
Specific Comments:
Comment 5: Line 35: “……indicated the water journal with a larger bubble had a greater impact with…”. I think that the phrase “with a larger bubble” is not necessary here in the Abstract, as the reader does not know the methodological tools for results evaluation yet. It is an information that concerns the results discussion.
Response 5: Authors agree to the concern raised by the reviewer. The sentence has been modified accordingly for proper understanding of the readers. Please see LN 35.
Comment 6: Line 74-75: “Researchers have sought to determine how much water should be extracted in order to have the least impact on the ecosystem (Koroneos et al., 2013).” The place of this sentence at this paragraph is not the appropriate because before this sentence authors presents the scope of the study as follow “In this context, the current study focuses on a bibliometric analysis of groundwater access and management to determine how far we have 72 progressed.” And after the sentence “Researchers have sought to determine how much water should be extracted in order to have the least impact on the ecosystem (Koroneos et al., 2013).” An extra presentation is occurred with more information. This interrupts the presentation of the study. So, the sentence “Researchers have sought to determine how much water should be extracted in order to have the least impact on the ecosystem (Koroneos et al., 2013).” ought to be moved.
Response 6: Authors agree to the concern raised by the reviewer. The paragraph has been modified accordingly. Please see LN 66-67 and 87-88.
Comment 7: Line 103: I believe that in the text there is no answer to the question h) Using the author's keywords and themes, what are the conceptual dynamics of groundwater research in India? Could you please mention it (if I miss something) or add it?
Response 7: Authors agree with the reviewer. The conceptual dynamics has been represented in the section 3.4
Comment 8: Lines 126-127: Did the authors try searches with other terms so as to gain higher number of results? e.g. instead of “access” to use “exploitation” or “extraction”.
Response 8: Authors agree to the concern raised by the reviewer. Here the theme of the study was the percentage of the population having access to groundwater. So after a brainstorming session with the authors, the keywords were selected for the manuscript.
Comment 9: Line 128: “…..and it spanned the years 1989 through 2022.” The authors declared in the Abstract that the period is from 1985 to 2022. Which is the correct period?
Response 9: The correct time period is 1985 to 2022. Correction has been incorporated in the line 142.
Comment 10: Lines 159: I do not believe that the statement “The figure depicts that there was an increasing trend of publications from 1985 to 2021……..” properly described the Figure 3. I believe that the increasing trend starts after 2000.
Response 10: Authors agree to the concern raised by the reviewer. The correction has been incorporated. Please see LN 173.
Comment 11: Line 212: A dot “.” is needed before the word Sandison as a new sentence must begin.
Response 11: As per the suggestion correction has been incorporated. Please see 231.
Comment 12: Lines 213-214: “…….piece of bibliographic data at the conclusion of a text ä send comments, footnotes, and so on a or retrieved from a citation index.” I do not understand the “ä” and the “a” in the phrase “…..and so on a or retrieved from a citation index”. These two “a” are added by mistake? If there is a reason, please explain.
Response 12: Authors agree to the concern raised by the reviewer. The correction has been incorporated. Please see 232.
Reviewer 2 Report
The paper describes an interesting approach to assessing research progress in the field of groundwater access and management. The study was based on the three terms Groundwater, Access and Management. Only articles that had these words in the title or abstract were considered relevant.
Authors: Please confirm or correct.
The reviewer is aware of many valuable articles on groundwater where the term Access or Management does not appear, particularly on groundwater processes such as recharge; some avoid the term Access and use Extraction or Abstraction. Others reject the fashionable term Groundwater Management because we cannot and should not manage the groundwater reservoir like water in the reservoir of a dam. The word manage is derived from the two Latin words manus (hand) and agere (to act). We can hardly handle (influence)crucial processes like groundwater recharge.
So searching for selected terms without considering others carries the risk that only researchers of the "same school" are evaluated and an imbalance arises.
Authors: Make a comment on it.
Author Response
Response to Reviewer
Comment 1: The paper describes an interesting approach to assessing research progress in the field of groundwater access and management. The study was based on the three terms Groundwater, Access and Management. Only articles that had these words in the title or abstract were considered relevant.
Authors: Please confirm or correct.
Response 1: Yes, only articles that had these words in the title or abstract were considered relevant.
Comment 2: The reviewer is aware of many valuable articles on groundwater where the term Access or Management does not appear, particularly on groundwater processes such as recharge; some avoid the term Access and use Extraction or Abstraction. Others reject the fashionable term Groundwater Management because we cannot and should not manage the groundwater reservoir like water in the reservoir of a dam. The word manage is derived from the two Latin words manus (hand) and agere (to act). We can hardly handle (influence) crucial processes like groundwater recharge.
So searching for selected terms without considering others carries the risk that only researchers of the "same school" are evaluated, and an imbalance arises.
Authors: Make a comment on it.
Response 2: We agree with the reviewer's point of view that other terms, such as extraction or abstraction, might have been used for such kinds of studies. However, the term 'groundwater' is common in all those studies. Based on the suggestion, when we performed a re-analysis, we got similar kinds of non-significant results. So it's a humble suggestion that the manuscript may be accepted in its present form. Moreover, we focused only on the Access of groundwater to the consumers and using other terms may deviate the focus of the study. Rest all the suggestions of both the reviewers are very well addressed and incorporated.
Round 2
Reviewer 2 Report
TThe article describes a method of limited significance because target words are chosen arbitrary, s. first review. However, the approach is principally interesting.